# Binding to DCAF1 distinguishes TASOR and SAMHD1 degradation by HIV-2 Vpx

**Michaël M. Martin**[ID]◉, **Roy Matkovic**[ID]◉, **Pauline Larrous**[ID], **Marina Morel**[ID], **Angélique Lasserre**[ID], **Virginie Vauthier**[ID], **Florence Margottin-Goguet**[ID]*

Institut Cochin, Université de Paris, INSERM U1016, Paris, France

◉ These authors contributed equally to this work.
* florence.margottin-goguet@inserm.fr

**Data Availability Statement:** All relevant data are within the manuscript and its Supporting Information files.

## Abstract

Human Immunodeficiency viruses type 1 and 2 (HIV-1 and HIV-2) succeed to evade host immune defenses by using their viral auxiliary proteins to antagonize host restriction factors. HIV-2/SIVsm Vpx is known for degrading SAMHD1, a factor impeding the reverse transcription. More recently, Vpx was also shown to counteract HUSH, a complex constituted of TASOR, MPP8 and periphilin, which blocks viral expression from the integrated viral DNA. In a classical ubiquitin ligase hijacking model, Vpx bridges the DCAF1 ubiquitin ligase substrate adaptor to SAMHD1, for subsequent ubiquitination and degradation. Here, we investigated whether the same mechanism is at stake for Vpx-mediated HUSH degradation. While we confirm that Vpx bridges SAMHD1 to DCAF1, we show that TASOR can interact with DCAF1 in the absence of Vpx. Nonetheless, this association was stabilized in the presence of Vpx, suggesting the existence of a ternary complex. The N-terminal PARP-*like* domain of TASOR is involved in DCAF1 binding, but not in Vpx binding. We also characterized a series of HIV-2 Vpx point mutants impaired in TASOR degradation, while still degrading SAMHD1. Vpx mutants ability to degrade TASOR correlated with their capacity to enhance HIV-1 minigenome expression as expected. Strikingly, several Vpx mutants impaired for TASOR degradation, but not for SAMHD1 degradation, had a reduced binding affinity for DCAF1, but not for TASOR. In macrophages, Vpx R34A-R42A and Vpx R42A-Q47A-V48A, strongly impaired in DCAF1, but not in TASOR binding, could not degrade TASOR, while being efficient in degrading SAMHD1. Altogether, our results highlight the central role of a robust Vpx-DCAF1 association to trigger TASOR degradation. We then propose a model in which Vpx interacts with both TASOR and DCAF1 to stabilize a TASOR-DCAF1 complex. Furthermore, our work identifies Vpx mutants enabling the study of HUSH restriction independently from SAMHD1 restriction in primary myeloid cells.

## Author summary

Human Immunodeficiency Virus (HIV) is still a major public health issue. The understanding of the molecular battle occurring during viral infection, between HIV components and cellular antiviral factors, the so-called restriction factors, is a key determinant

**Funding:** FMG received grants from the "Agence Nationale de la Recherche sur le SIDA et les hépatites virales" (ANRS) and SIDACTION to support this work. FMG is supported by INSERM and MM by CNRS. MMM was supported by SIDACTION; RM by ANRS and SIDACTION; VV by ANRS. PL and AL received a fellowship from the French government. The funders had no role in study design, data collection and analysis, decision to publish, or preparation of the manuscript.

**Competing interests:** The authors have declared that no competing interests exist.

for new treatment development. Namely, HIV auxiliary proteins are powerful to induce the downregulation of cellular restriction factors by hijacking the Ubiquitin/proteasome pathway, in order to facilitate the completion of a well-processed HIV replication cycle. For instance, HIV-2 Vpx eases reverse transcription in myeloid cells by counteracting the SAMDH1 restriction factor. More recently, we discovered the ability of Vpx to induce the degradation of the HUSH epigenetic repressor complex to favor in turn, the expression of the provirus. In this study, we uncovered the mechanisms by which Vpx antagonizes TASOR, the core subunit of the HUSH complex. We highlighted key differences between Vpx-induced TASOR and SAMHD1 degradation. These findings will help to propose strategies to study or to target either HUSH or SAMHD1, especially in myeloid cells where SAMHD1 restriction operates.

## Introduction

Human Immunodeficiency viruses type 1 and 2 (HIV-1 and HIV-2), responsible for Acquired Immunodeficiency Syndrome (AIDS), appeared in humans after cross-species transmission of non-human primate viruses (Simian Immunodeficiency viruses or SIV). They encode for viral auxiliary proteins, which play a major role in helping the virus to evade hurdles represented by "Restriction factors" [1]. Commonly, they act as viral antagonists engaging specific ubiquitin ligases to induce the ubiquitination and subsequent degradation of the restriction factor and, in turn, enabling the virus to bypass a specific block along the viral life cycle [2]. The Cullin-RING-type class of E3 ubiquitin ligases, constituted of a central Cullin scaffold protein and a catalytic RING subunit [3], represent the major class of ubiquitin ligases hijacked by lentiviral proteins including Vpr and Vpx [2]. All extant lentiviruses encode for Vpr, while Vpx is only encoded by the HIV-2/SIVmac/SIVsmm (infecting human, macaque and sooty mangabey, respectively) and SIVrcm/mnd-2 (infecting red-capped mangabey and mandrill) lineages [4–6]. Both Vpr and Vpx are incorporated into virions [7,8] and present structural (3α-helices and unstructured N- and C-termini tails) and functional similarities, likely due to an ancestral Vpr gene duplication or a recombination event, which has given rise to Vpx [4–6,9]. To ensure their functions, both viral proteins bind to the DDB1- and Cul4A-associated factor 1 (DCAF1) substrate adaptor of the host Cullin4A-RING ubiquitin ligase [10–19]. While Vpr exhibits a wide range of cellular substrates [20–28], reviewed in [29], Vpx seems to target only a few pathways *via* DCAF1 [30–33]. At the forefront, HIV-2/SIVsmm Vpx induces the degradation of SAMHD1, relieving a block at the reverse transcription step [31,32]. SAMHD1 is a deoxy-nucleotide triphosphate (dNTP) triphosphohydrolase that restricts HIV replication by lowering the pool of dNTP and thereby inhibits the synthesis of viral DNA in non-dividing cells, macrophages and quiescent CD4+ T cells [34–38]. Vpx bridges SAMHD1 to DCAF1, leading to SAMHD1 poly-ubiquitination and subsequent degradation [31,32,39]. Intriguingly, Vpx/Vpr proteins from divergent viruses, despite their structural homology, target entirely different domains of SAMHD1 in a species-specific manner [40–42]. This difference in SAMHD1 recognition is evolutionarily dynamic and is further witnessed by sites of positive selection in both N- and C-terminal domains of the host protein [9,40,43]. Crystal structures of SIVsmm Vpx, the C-terminal WD40 domain of DCAF1 and the C-terminal region of human SAMHD1 complex, and between SIVmnd-2 Vpx, the C-terminal WD40 domain of DCAF1 and the N-terminal region of mandrill SAMHD1 highlight a conserved mode of interaction of Vpx with DCAF1, as well as providing clues as to how a conserved DCAF1-Vpx module can bind different SAMHD1 from different host-species [41,42]. In addition to SAMHD1, we and others

uncovered the ability of HIV-2/SIVsmm Vpx to induce the degradation of HUSH, an epigenetic complex repressing the expression of transgenes, retroelements and hundreds of cellular genes [30,33,44–47]. The HUSH complex is constituted of three subunits, TASOR, MPP8 and Periphilin, with TASOR acting as a core member by interacting with both MPP8 and periphilin [47,48]. TASOR contains at its N-terminus part an inactive poly ADP-ribose polymerase (PARP)-*like* domain essential for HUSH-mediated repression [48].

By degrading HUSH, Vpx favours viral expression in a model of HIV-1 latency, which is referred to "viral reactivation" thereafter [30,33]. Altogether, Vpx induces the degradation of two antiviral proteins acting at two different steps of the viral life cycle: SAMHD1, at the reverse transcription step, and HUSH, at a post-integration step.

As of today, the molecular determinants of SIVsmm Vpx involved in SAMHD1 antagonism are well characterized. Indeed, Vpx interacts directly with SAMHD1 and DCAF1, with residues located in the α-helices 1 and 3 for SAMHD1 and several residues along Vpx for DCAF1 as described in [5,39,42]. However, the molecular details of how Vpx promotes HUSH degradation have been poorly investigated. Our previous results suggest that HUSH destabilization by Vpx occurs within a TASOR-MPP8 complex, with TASOR being the most efficiently impacted by Vpx. Several results initially led us to hypothesize that the mechanism of HUSH degradation by Vpx is likely to follow the mechanism of SAMHD1 degradation. Indeed, we showed that DCAF1 depletion or mutation of the Q76 residue in SIVsmm Vpx, critical for binding to DCAF1 [14], prevented TASOR degradation. In addition, SIVsmm Vpx Q76R interacted with TASOR. On the other hand, SIVsmm Vpx Q47A-V48A was identified as a TASOR-binding-deficient mutant that failed to induce TASOR degradation, but still degraded SAMHD1. Altogether these results suggested a "ubiquitin ligase hijacking model", in which Vpx would bridge DCAF1 to the HUSH complex. Here we have challenged this model by performing additional interaction and functional experiments and by characterizing new HIV-2 Vpx mutants. Our results demonstrate that TASOR binds DCAF1 independently from Vpx, while Vpx reinforces the strength of interaction between the two cellular proteins. Furthermore, the ability of Vpx to interact with DCAF1 correlates with TASOR degradation by Vpx, while degradation of SAMHD1 is efficient even when binding of Vpx to DCAF1 is strongly impaired. These results led us to speculate that a novel rearrangement of the HUSH-DCAF1 interaction by Vpx is necessary for HUSH degradation.

## Results

### TASOR interacts with DCAF1 and this association is stabilized by HIV-2 Vpx

The study here is dedicated to HIV-2 Vpx from the Ghana-1 strain [49]. In agreement with our previous results obtained using SIVsmm Vpx proteins [30], immunoprecipitation of HA-tagged Vpx from HIV-2 with anti-HA antibodies pulled-down TASOR-Flag (Fig 1A, lane 5). The DCAF1 binding-deficient Vpx Q76R mutant has kept its ability to bind TASOR (Fig 1A, lane 6). Of note, HIV-2 HA-Vpx was detected under two forms of different molecular weight, at about 15-kDa and 30-kDa (Fig 1A or 1B, input). The 15-kDa form corresponds to the expected size of monomeric Vpx, while the 30-kDa form is likely a dimer of the protein. This 30-kDa form is non-denaturable, neither by DTT or β-mercaptoethanol reducing agents nor by 8M Urea-containing lysis buffer as previously characterized in [50] (S1A Fig). Both the 15-kDa and 30-kDa forms were recognized by anti-Vpx antibodies (S1B Fig). The 30-kDa Vpx form was better immunoprecipitated with TASOR than the monomeric form following anti-Flag-mediated immunoprecipitation of TASOR-Flag (Fig 1B, lane 6). Nonetheless, the interaction was confirmed between Flag-tagged Vpx, which is unable to form an apparent dimer, and

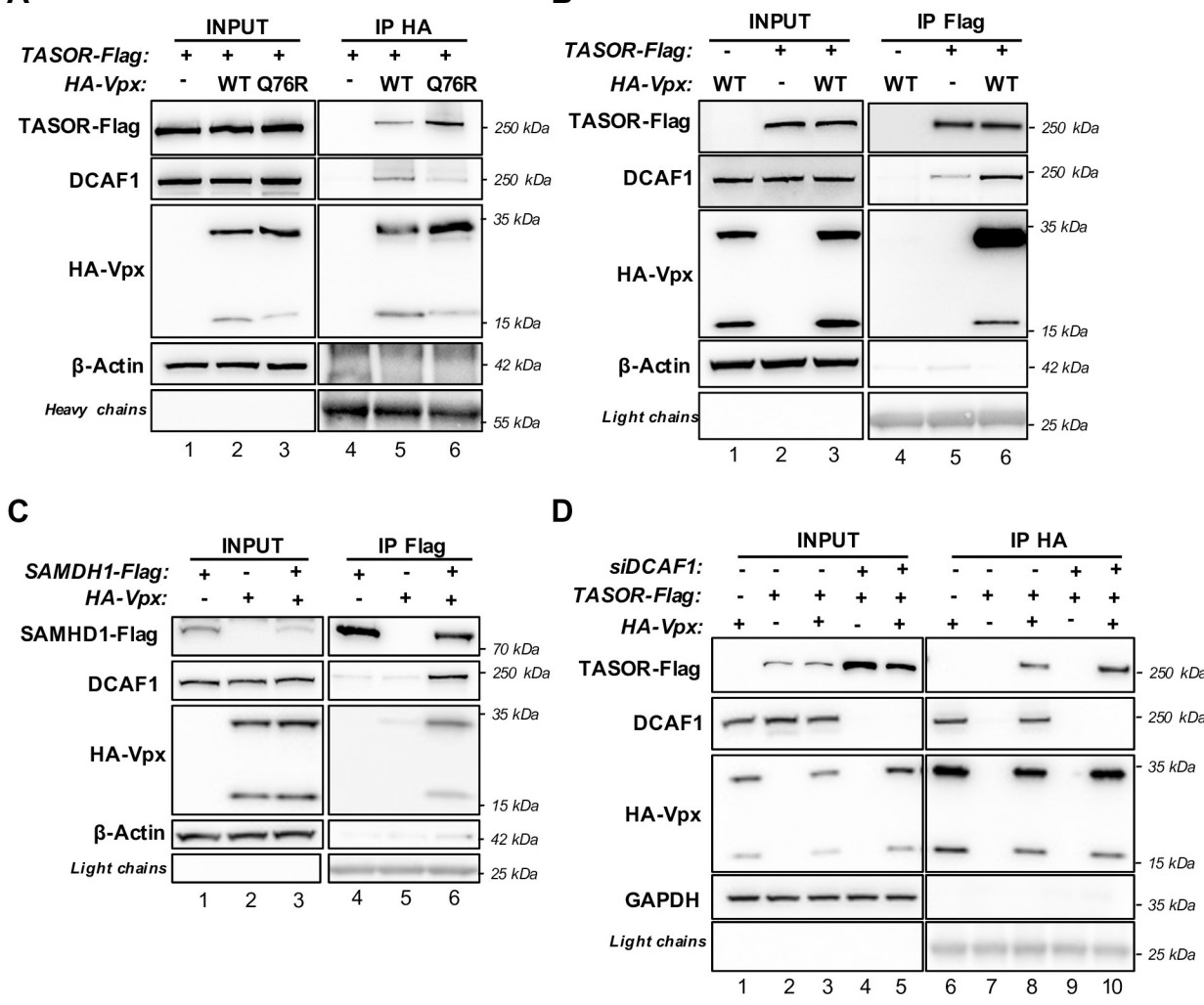

**Fig 1. Interaction between TASOR and DCAF1 is stabilized in the presence of Vpx.** (**A**) HA-Vpx WT or DCAF1 binding-deficient HA-Vpx Q76R proteins were co-expressed with TASOR-Flag in HeLa cells, then an anti-HA immunoprecipitation was performed. (**B**) TASOR-Flag and HA-Vpx WT were co-expressed in HeLa cells, then an anti-Flag immunoprecipitation was performed. (**C**) SAMHD1-Flag and HA-Vpx WT were co-expressed in HeLa cells, then an anti-Flag immunoprecipitation was performed. (**D**) HeLa cells were treated with either siRNA CTL or siRNA DCAF1. After 24h, HA-Vpx WT and TASOR-Flag were co-expressed for 48h, then an anti-HA immunoprecipitation was performed. In each panel, the indicated proteins were revealed by western blot. For this figure, each shown immunoblot is representative of at least 3 independent experiments.

HA-TASOR, suggesting that monomeric Vpx binds TASOR efficiently (S1C Fig). To our surprise, endogenous DCAF1 was immunoprecipitated with TASOR-Flag in the absence of Vpx (Fig 1B, lane 5), though this TASOR-DCAF1 interaction was reinforced in the presence of Vpx (Fig 1B, compare lanes 5 and 6, DCAF1 panel). In contrast, DCAF1 was immunoprecipitated with SAMHD1-Flag only in the presence of Vpx as expected [39] (Fig 1C, lane 6). Interaction between TASOR and DCAF1 isoform 1 or DCAF1 isoform 3, which lacks the chromodomain present in isoform 1 [51], was confirmed following overexpression of the proteins (S2A Fig). Thus, while Vpx bridges SAMHD1 to DCAF1, TASOR interacts with DCAF1 independently from Vpx. This result raised the possibility that DCAF1, as an adaptor of a ubiquitin ligase complex, could be involved in TASOR turnover. Nonetheless, in our conditions, we could not observe any endogenous TASOR levels modulation upon DCAF1 depletion (S2B Fig). We

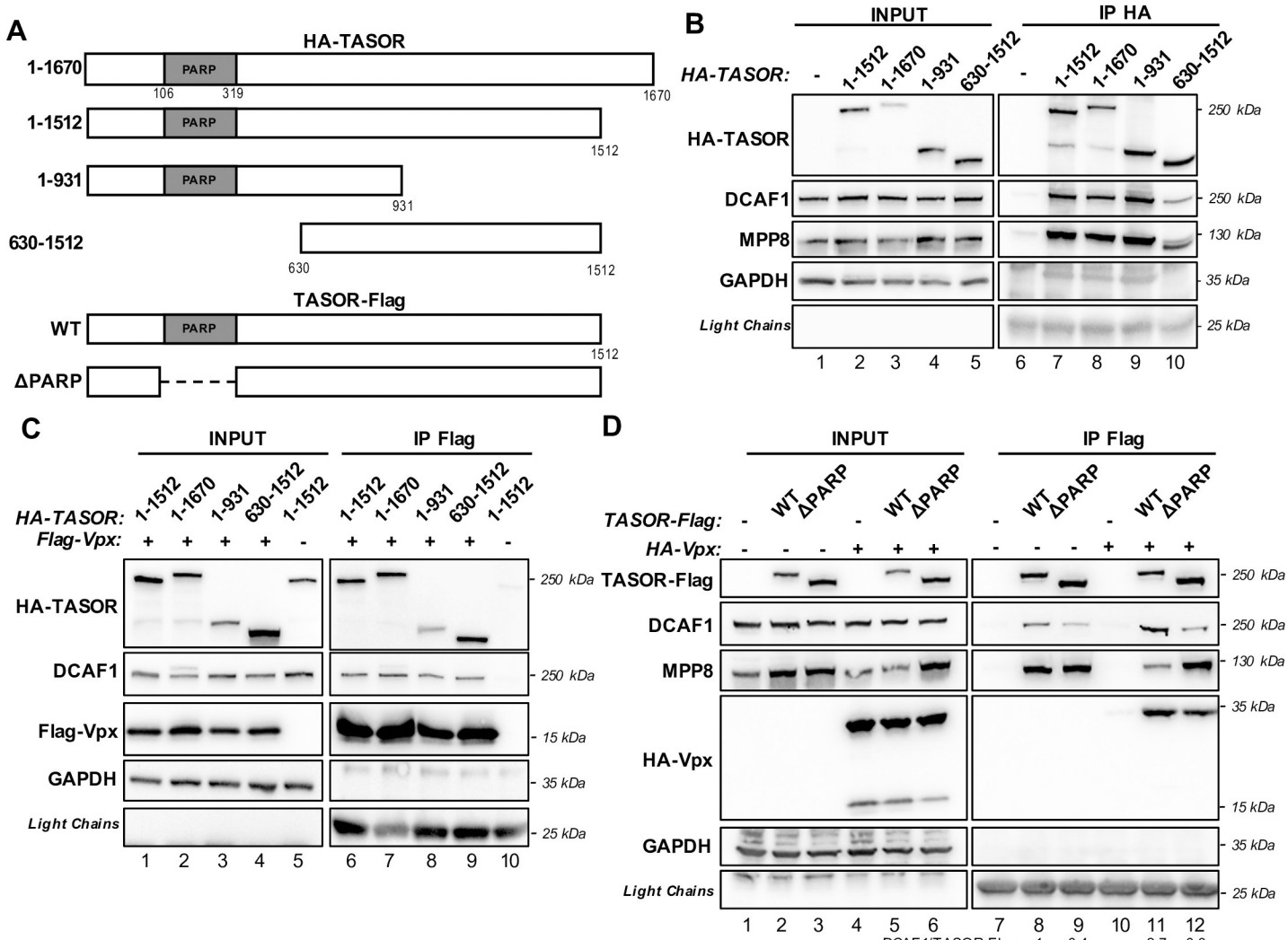

**Fig 2. TASOR PARP-*like* domain is involved in DCAF1 binding.** (**A**) Schematic representation of HA-tagged TASOR or Flag-tagged TASOR constructs. *1–1670*: TASOR long isoform. *1–1512*: TASOR short isoform. *1–931*: N-terminal fragment of TASOR. *630–1512*: C-terminal fragment of TASOR short isoform. *WT*: TASOR (short isoform). *ΔPARP*: TASOR (short isoform) deleted of its PARP-*like* domain (106–319 aa). (**B**) Indicated HA-TASOR constructions were expressed in HeLa cells, then an anti-HA immunoprecipitation was performed. (**C**) Flag-Vpx WT and indicated HA-TASOR constructs were co-expressed in HeLa cells, then an anti-Flag immunoprecipitation was performed. (**D**) TASOR-Flag WT or TASOR-ΔPARP-Flag were co-expressed with HA-Vpx WT in HeLa cells, then an anti-Flag immunoprecipitation was performed. In each panel, the indicated proteins were revealed by western blot. For this figure, each shown immunoblot is representative of at least 3 independent experiments.

further wondered whether binding of Vpx to TASOR could depend on a pre-existing TASOR-DCAF1 complex. The interaction between TASOR-Flag and HA-Vpx or between HA-TASOR and Flag-Vpx was unaffected by inhibition of DCAF1 expression (Figs 1D, compare lanes 10 to 8 and S2C). These results, together with the finding of an association between Vpx Q76R and TASOR, suggest that Vpx can interact with TASOR independently from DCAF1.

Because Vpx stabilizes the TASOR-DCAF1 interaction, we expected Vpx and DCAF1 to bind distinct domains of TASOR. Fig 2A depicts the TASOR constructs we tested for their interaction with DCAF1: 1–1670 and 1–1512 represent two distinct TASOR isoforms that differ only in their C-terminal part, with 1–1512 being the one we have used in this manuscript.

DCAF1 interacted preferentially with the 930 first amino acids of TASOR (Fig 2B, compare lanes 9 and 10). In contrast to DCAF1, Vpx perfectly binds the 630–1512 C-terminal fragment of TASOR (Fig 2C). The N-terminus part of TASOR contains the PARP-*like* domain with no catalytic activity, but required to maintain transgene repression by HUSH [48]. Depletion of this domain reduced binding of DCAF1 to TASOR both in the absence or in the presence of Vpx (Fig 2D, compare lanes 9 to 8 and 12 to 11). In contrast, TASOR-ΔPARP was still able to bind equivalently to MPP8 in the absence of Vpx in agreement with results from Douse et al. [48] (Fig 2D, lanes 8,9), supporting the fact that the binding affinity of DCAF1 to TASOR-ΔPARP is weaker. Both TASOR WT and TASOR-ΔPARP over-expressions tend to increase MPP8 protein levels (Fig 2D, lanes 1–3). However, upon Vpx expression, endogenous MPP8 accumulation could only be observed with the PARP-*like* truncated version of TASOR and not with TASOR WT, suggesting that TASOR N-terminus part is necessary for Vpx-mediated HUSH degradation (Fig 2D, lanes 4–6 compared to lanes 1–3). Thereby, these results suggest that DCAF1 binds the N-terminus part of TASOR. Of importance, Vpx binds TASOR-ΔPARP as well as TASOR-WT (Fig 2D, lanes 11,12) suggesting that DCAF1 and Vpx interact with different regions in TASOR. However, the TASOR-DCAF1 interaction was no more stabilized by Vpx when TASOR PARP-*like* domain was removed (Fig 2D, compare lanes 11 to 8, 12 to 9, quantification under the Figure).

Altogether, our results suggest the existence of a ternary complex between TASOR, DCAF1 and Vpx with independent possible binary interactions between the three partners.

## Vpx Q47AV48A and Vpx V48A have an unexpected defect in DCAF1 binding

To further study the link between TASOR binding, TASOR degradation and increase of HIV-1 minigenome expression, we investigated the impact of the individual Q47A and V48A mutations in HIV-2 Vpx, keeping in mind that we previously found SIVsmm Vpx Q47A-V48A to be impaired in TASOR binding and degradation [30]. Vpx proteins (WT or mutants) were incorporated into Viral-*Like*-Particles (VLPs). Their incorporations were checked and adjusted in order to deliver about the same quantity of viral proteins into J-Lat A1 T cells [52] (S3A Fig). These cells harbor a latent HIV-1-LTR-Tat-*IRES*-GFP-LTR cassette, which GFP expression is being cooperatively increased upon Vpx-mediated TASOR degradation along with Tumor Necrosis Factor alpha treatment (TNFα) [30]. As in SIVsmm Vpx, mutation of both Q47 and V48 residues to alanines (QV mutant) in HIV-2 Vpx, strongly, but not fully, impaired TASOR degradation and viral reactivation (Fig 3A and 3B). Because J-Lat T cells do not express SAMHD1, we analyzed SAMHD1 degradation in the THP-1 myeloid cell line. The QV mutant was able to induce SAMHD1 degradation (Fig 3C). Vpx V48A, alike Vpx QV, was impaired in both TASOR degradation and viral reactivation, but not in SAMHD1 degradation. In contrast, Vpx Q47A could degrade both TASOR and SAMHD1, and was able to increase the GFP expression derived from the integrated HIV-1 minigenome (Fig 3A, 3B and 3C). Therefore, the V48A mutation is at stake in the loss of HUSH antagonism by the Vpx Q47A-V48A mutant.

Interaction experiments were further undertaken with proteins expressed from transfected vectors. Of note, the three HA-tagged Vpx mutants, Q47A, V48A and QV, were detected as monomers, but did not produce the 30-kDa form (Fig 3D, input, left). Following an anti-HA (Vpx) immunoprecipitation, none of them were found in association with TASOR, in contrast to WT Vpx (Fig 3D, TASOR-Flag panel). As this was inconsistent with Vpx Q47A being functional as well as WT Vpx (Fig 3A and 3B), we performed the reverse co-immunoprecipitation experiment pulling down first TASOR-Flag (Fig 3E). In these conditions, we could detect an

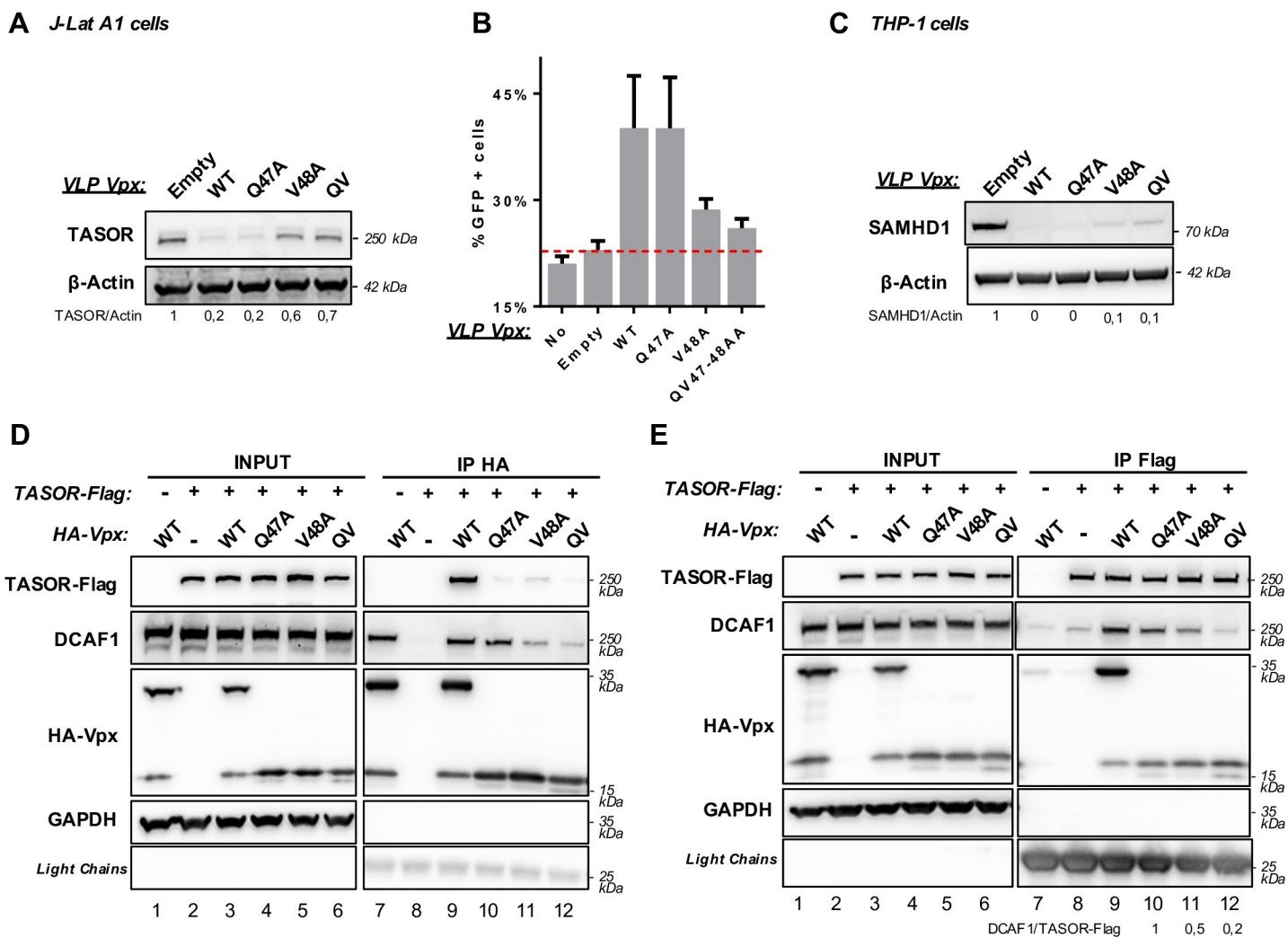

**Fig 3. Vpx Q47AV48A loss of activity in J-Lat A1 T cells results from the V48A mutation.** (**A** and **B**) HIV-2.Gh1 Vpx WT or indicated mutants were tested for TASOR degradation (**A**) and viral reactivation in J-Lat A1 T cells (**B**). J-Lat A1 T cells were treated with Vpx-containing VLPs. After overnight treatment with TNF-α, cells were analyzed by flow cytometry and whole-cell extracts by western blot. For each mutant, reactivation assay was performed at least 3 times and the immunoblot shown is representative of at least 3 independent VLP productions, SD is shown. (**C**) HIV-2.Gh1 Vpx WT or indicated mutants were tested for SAMDH1 degradation. Non-differentiated THP-1 cells were treated for 24h with VLPs and whole-cell extracts were analyzed by western blot. The shown immunoblot is representative of at least 3 independent VLP productions. *Empty*: VLP in which Vpx is not incorporated. *QV*: Vpx Q47A-V48A double mutant. (**D** and **E**) HA-Vpx WT or indicated mutants were co-expressed along with TASOR-Flag in HeLa cells, then an anti-HA (**D**) or anti-Flag (**E**) immunoprecipitation was performed. shown immunoblots are representative of 3 independent experiments.

interaction between TASOR and monomeric Vpx proteins, including Vpx Q47A (Fig 3E). However, Vpx V48A and the QV mutant did not promote TASOR-DCAF1 interaction as well as Vpx Q47A, in agreement with the functional data (Fig 3E, DCAF1 panel and quantification under the Figure). Thus, we also tested the ability of these Vpx mutants to bind DCAF1. In our previous study [30], we assumed Vpx Q47A-V48A would bind DCAF1 as it was able to degrade SAMHD1. However, to our surprise, interaction affinity of Vpx Q47A-V48A or Vpx V48A to DCAF1 was strongly reduced, while Vpx Q47A could bind DCAF1 as WT Vpx (Fig 3D, DCAF1 panel). Similar results were obtained with HA-tagged SIVsmm Vpx mutated on Q47 and V48 residues (S4A Fig, lanes 12–14) and with HIV-2 Flag-Vpx, which is fully active for TASOR degradation (S4B Fig) and which does not form an apparent dimer (S4C Fig,

interaction with DCAF1, lanes 14 and 15, and S4D Fig, interaction with TASOR, lanes 14 and 15). Hence, the loss of activity of Vpx Q47A-V48A or Vpx V48A could result from lower binding affinity to DCAF1 and not necessarily from a loss of binding to TASOR.

## Integrity of a set of Vpx exposed residues is required for HUSH antagonism

In an attempt to map Vpx residues important for TASOR binding, we put forward the hypothesis that these residues would not be in contact with DCAF1 or SAMHD1 (see Fig 4A for such contacts). SIVsmm Vpx lacking the C-terminal poly-proline tail is still able to assemble into a ternary complex with SAMHD1 and DCAF1 and to induce SAMHD1 ubiquitination [39,53]. Thus, we first focused on this flexible C-terminal tail but found that the apparent dimeric Vpx, deleted of this tail, could still interact with TASOR (S5 Fig). We then paid attention to exposed residues alike Vpx R42, which is present in most Vpx from the HIV-2/SIVsmm lineage and Vpr proteins from the SIVagm lineage able to counteract HUSH, but not in some lineages defective for HUSH degradation [30]. When looking at the published crystal structure of the complex between SIVsmm Vpx, the C-terminus domains of DCAF1 and SAMHD1, the residue R42 appeared accessible, i.e. in contact neither with SAMHD1 nor with DCAF1, and located in a charged area with several Arginine and Acid glutamic residues (Fig 4A, based on [42]). Therefore, we decided to substitute R42 and these other charged residues, namely E30, R34 from α-helix 1; E43, R51, R54 and D58 from α -helix 2. In this second helix, we also mutated other exposed residues alike V37, N38, F46, W49, Q50, and did modify the C89 and L90 residues in the C-terminal tail (Fig 4A). All mutants were tested for TASOR and SAMHD1 degradation in J-Lat A1 T and THP-1 cells (Fig 4B and 4C). The quantity of VLP was adjusted to transduce the same quantity of Vpx proteins (S3A Fig). Some Vpx mutants were impaired in the degradation of both TASOR and SAMHD1 (W49A, C89A, D58A) and were not further considered. Among mutations that do not impact SAMHD1 degradation, some (R42A, V37A, N38A or L90A and R42A-QV (RQV) triple mutant) strongly impaired TASOR degradation and reactivation in J-Lat A1 T cells, while others (R34A, E43A, F46A, R51A) only reduced TASOR degradation and viral reactivation. Other Vpx mutants had no impact (E30A, Q50A, R54A). Altogether, our results suggest that the integrity of a set of Vpx exposed residues (residues in red on Fig 4A) is important for HUSH antagonism. Importantly, a good correlation was obtained between Vpx ability to degrade TASOR and Vpx-mediated reactivation in J-Lat A1 T cells (Fig 4B).

## Vpx mutants defective for TASOR but not SAMHD1 degradation show a defect in DCAF1 binding

Strikingly, all the Vpx mutants proficient for SAMHD1 degradation, but partially or totally deficient for TASOR degradation, were able to bind TASOR (S6A and S6B Fig), except monomeric V48A, as shown in Fig 3D, and RQV mutants, when pulling down first HA-Vpx. In contrast, we noticed that some of them, alike R34A, R42A, R51A, bound less efficiently to DCAF1 (S6A Fig). The reduced binding affinity of Vpx R34A and Vpx R42A for DCAF1 was reproducible (Fig 5A and 5B). The double mutant Vpx R34A-R42A (RR) showed a dramatic decrease of its affinity for DCAF1, while it was still interacting with TASOR (Fig 5A, lane 12 and 5B). Again, similar results were obtained with SIVsmm Vpx mutated on R42 residue (S4A Fig, lane 11) and with HIV-2 Flag-Vpx mutants (S4B Fig, TASOR degradation, S4C, interaction with DCAF1, lanes 10–13, and S4D, interaction with TASOR, lanes 10–13). In addition, the interaction between TASOR-Flag and DCAF1 was not stabilized by Vpx RR or Vpx R42A (Fig 5C, lanes 11 and 12, DCAF1 panel). In turn, Vpx R42A and Vpx RR could neither degrade TASOR, nor reactivate HIV-1 in J-Lat-A1 cells, while they could degrade SAMHD1 in THP-1

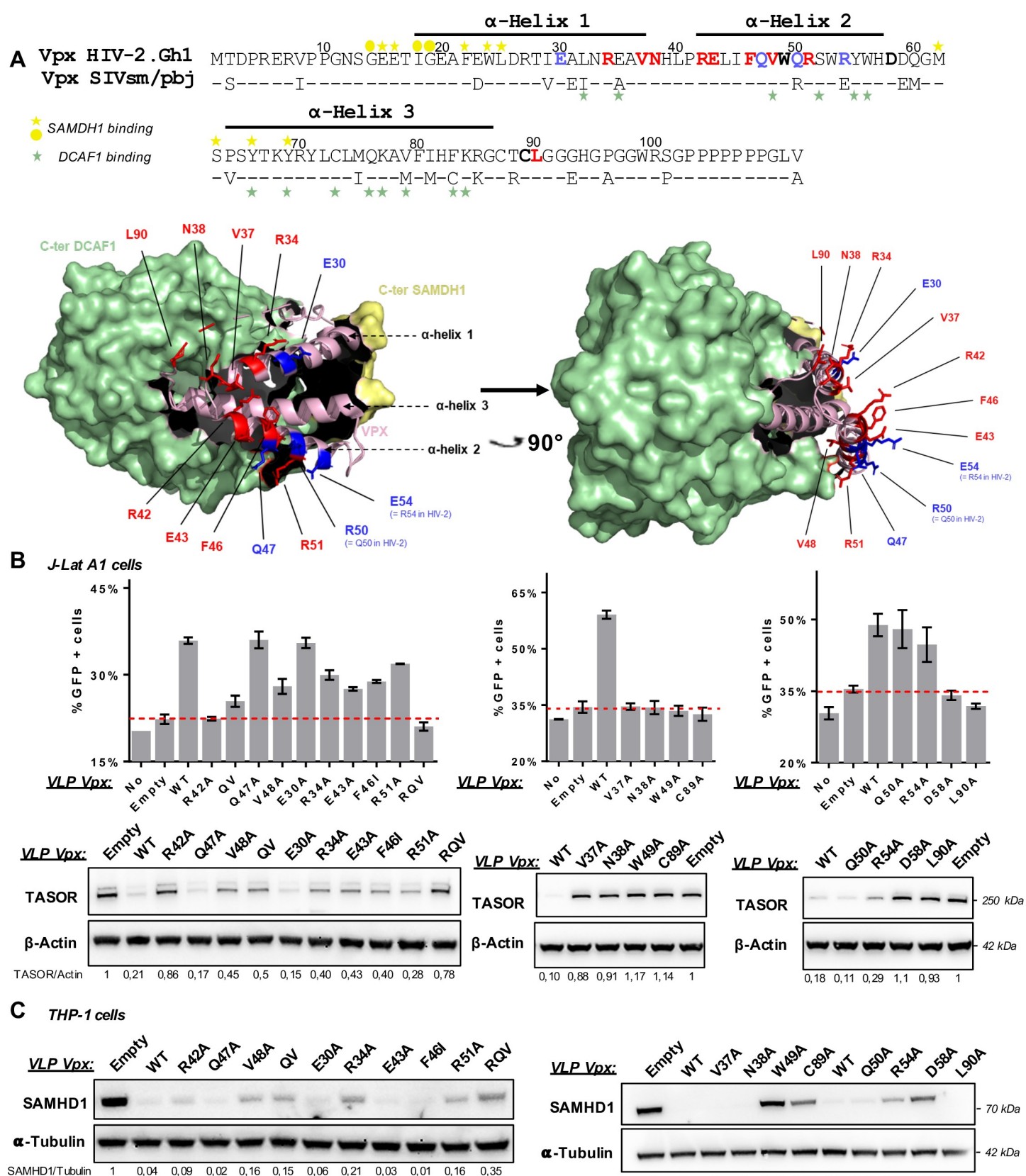

**Fig 4. The integrity of a set of Vpx exposed residues from α-helix 1 and 2 and the C-ter tail is required for HUSH antagonism.** (**A**) The representation of the crystal structure of CtD-huDCAF1/Vpx SIVsmm/CtD-huSAMDH1 ternary complex, resolved by Schwefel *et al* in 2014 [42] (PDB: 4CC9), has been adapted here to highlight Vpx residues that we have tested in this study regarding SAMHD1 and TASOR degradations. *Top*: sequence alignment of HIV-2.Gh1 Vpx and SIVsmm Vpx. Our study is dedicated to HIV-2 Vpx, while the structure was obtained with SIVsmm Vpx. The two sequences share 79.81% identity and 92% similarity. Green and Yellow marks indicate residues of SIVsmm Vpx involved in DCAF1 and SAMDH1 binding respectively, according to Schwefel *et al.*[42] *Stars*: interaction mediated by the lateral chain. *Dot*: Interaction mediated by the principal chain. The integrity of residues shown in red is important for HUSH degradation, while this is not the case for residues in blue. Residues in bold dark are important for HUSH and SAMDH1 degradation. *Bottom*: Two different views of the CtD-huDCAF1/Vpx SIVsmm/CtD-huSAMHD1 complex, resolved by Schwefel *et al* in 2014[42] (PDB: 4CC9). The huDCAF1 and huSAMHD1 C-terminal domains are shown in green and yellow, respectively. Vpx SIVsmm is shown in light pink. Tested residues involved (red) or not (blue) in HUSH antagonism are indicated. (**B**) HIV-2.Gh1 Vpx WT and mutants were tested for TASOR degradation and viral reactivation. J-Lat A1 T cells were treated with VLPs. After overnight treatment with TNF-α, HIV-1 LTR-driven GFP expression was analyzed by flow cytometry and whole-cell extracts by western blot. For each mutant, reactivation assay was performed 2, 3 or more times and shown immunoblots are representative of at least 2 independent experiments. *Empty*: VLP in which Vpx is not incorporated. *QV*: Vpx Q47A-V48A double mutant. *RQV*: Vpx R42A-Q47A-V48A triple mutant. (**C**) HIV-2.Gh1 Vpx WT and mutants were tested for SAMDH1 degradation. Non-differentiated THP-1 cells were treated 24h with VLPs and whole-cell extracts were analyzed by western blot. For each mutant, shown immunoblots are representative of 2 independent experiments.

cells, though a little less efficiently for the RR mutant (Figs 5D, 5E, 5F and S3B for VLPs incorporation). Of note, Vpx R34A showed a reduced binding to DCAF1, but still retained some ability to induce TASOR degradation and HIV-1 reactivation. Altogether, these observations suggest the necessity of a strong binding affinity between Vpx and DCAF1 to stabilize the interaction between TASOR and DCAF1 and to induce TASOR degradation. Such strong binding would not be as much requested for SAMHD1 degradation.

## Vpx RR and Vpx RQV degrade SAMHD1 but not TASOR in macrophages

Up to now, TASOR degradation was analyzed in the J-Lat A1 T-cell line and SAMHD1 degradation in the THP-1 myeloid cell line. Because degradation of one protein could impact degradation of the other, we questioned the phenotype of Vpx mutants in primary macrophages, in which SAMHD1 restriction operates. In macrophages, Vpx-mediated SAMHD1 depletion was very efficient: SAMHD1 was still not detected 7 days after Vpx delivery (Fig 6A). In contrast, TASOR protein levels reappeared at day 1 or day 2 following Vpx addition, depending on the efficacy of Vpx delivery by VLPs (Figs 6A and S3C for VLPs incorporation). In consequence, SAMHD1 and TASOR degradation by Vpx and mutants were monitored between 0 and 24 hours following Vpx addition. While all mutants efficiently induced SAMHD1 degradation, differences were observed regarding TASOR degradation. Vpx QV, Vpx Q47A, Vpx V48A and Vpx R34A were all able to degrade TASOR in macrophages, even if degradation was slightly less efficient with Vpx V48A and Vpx Q47A-V48A for some donors (Figs 6B and S7A). Vpx R42A could also induce TASOR degradation, but less efficiently (Figs 6B and S7A). Only Vpx RR and Vpx RQV were strongly impaired in TASOR degradation (Figs 6B and S7A). As a control, the Vpr protein from Vervet African Green monkey SIV induced TASOR but not SAMHD1 degradation as expected (S7B Fig) [9,30,33]. Altogether, two Vpx mutants (Vpx RR and Vpx RQV), characterized by a decreased affinity for DCAF1 binding, are no more able to degrade TASOR but are proficient for SAMHD1 degradation in macrophages.

## Discussion

The mechanism of Vpx-mediated degradation of HUSH relies on the use of the DCAF1 ubiquitin ligase adaptor suggesting the existence of a classical ubiquitin ligase hijacking model, in which Vpx would bridge HUSH to DCAF1, as it is the case for SAMHD1. Nonetheless, our findings here suggest that HUSH and SAMHD1 mechanisms present notable differences. Indeed, firstly, TASOR can interact with DCAF1, thanks to its PARP-*like* domain located in its N-terminal part, in the absence of Vpx, while SAMHD1 interacts with DCAF1 only in the presence of Vpx; secondly, the Vpx-mediated degradation of TASOR is less efficient than that of SAMHD1 (in agreement with results from our SILAC screen published in [30]); thirdly, an

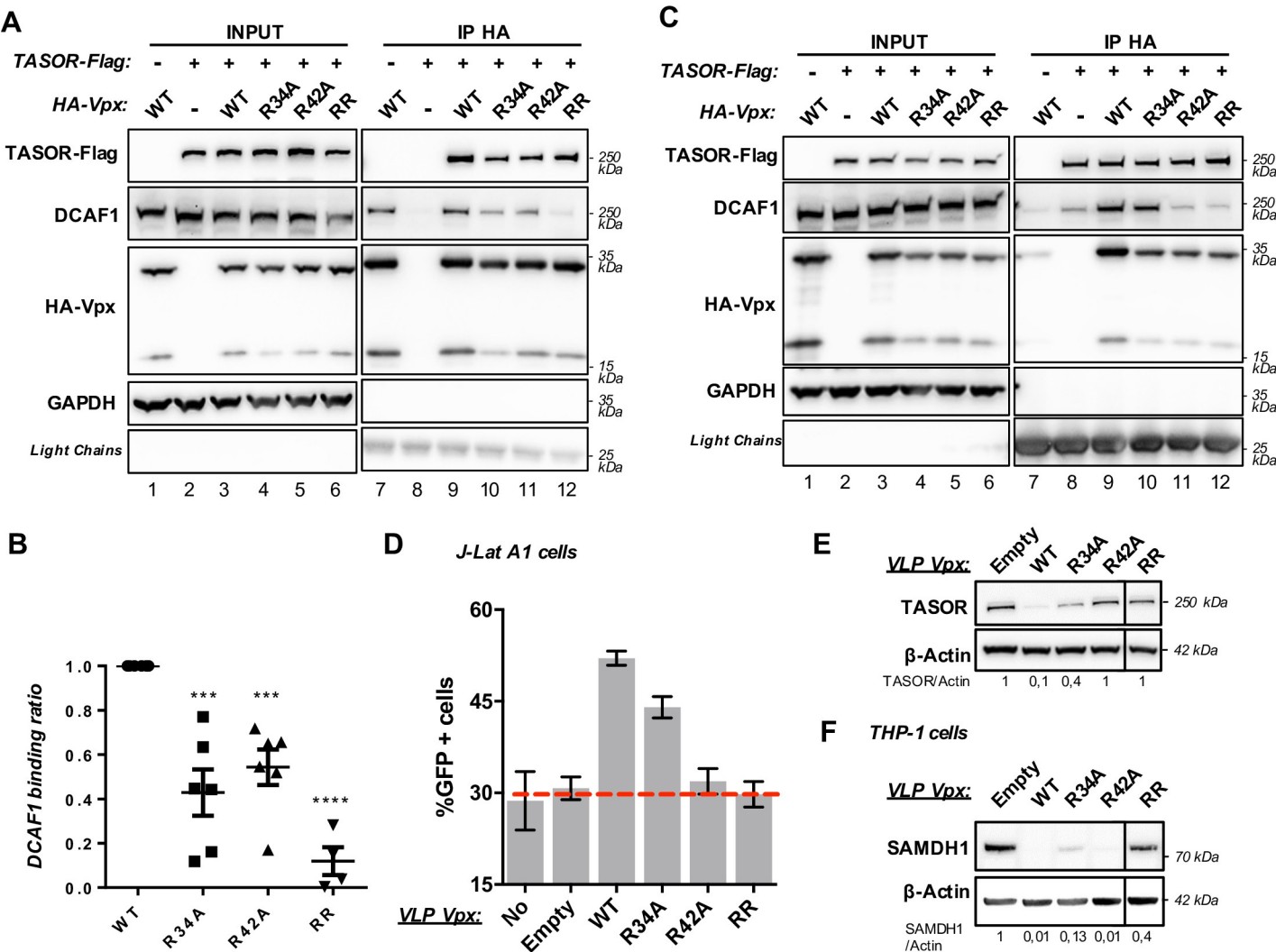

**Fig 5. Vpx R34A-R42A, which induces SAMHD1 but not TASOR degradation, is characterized by a reduced binding affinity for DCAF1.** (**A**) HA-Vpx WT or indicated mutants and TASOR-Flag were co-expressed in HeLa cells, then an anti-HA immunoprecipitation was performed. (**B**) Graph representing Vpx binding affinity to DCAF1. Co-immunoprecipitated DCAF1 and immunoprecipitated HA-Vpx (Vpx WT (n = 6), Vpx R34A (n = 6), R42A (n = 6) and RR (n = 4)) were quantified from independent experiments. Co-immunoprecipitated DCAF1 signal over immunoprecipitated Vpx signal ratios were calculated and reported to Vpx WT (ratio 1). (**C**) TASOR-Flag and HA-Vpx WT or indicated mutants were co-expressed in HeLa cells, then an anti-Flag immunoprecipitation was performed. Immunoblot is representative of at least 3 independent experiments. (**D** and **E**) HIV-2.Gh1 Vpx WT and mutants were tested for viral reactivation (**D**) and TASOR degradation (**E**). J-Lat A1 T cells were treated with VLPs. After overnight treatment with TNF-α, HIV-1-LTR-driven GFP expression was analyzed by flow cytometry and whole-cell extracts were analyzed by western blot. For each mutant, reactivation assay was tested at least 3 times (SEM is shown), and immunoblots are representative of at least 3 independent experiments. (**F**) HIV-2.Gh1 Vpx WT and mutants were tested for SAMDH1 degradation, THP-1 cells were treated with VLP overnight and then whole-cell extracts were analyzed by western blot. *RR*: Vpx double mutant R34A-R42A. For each mutant, immunoblots are representative of 3 independent experiments.

apparently weaker interaction between Vpx and DCAF1 has no impact on the efficiency of SAMHD1 degradation, while it seems critical to stabilize the TASOR-DCAF1 interaction and thus the degradation of TASOR (model Fig 7).

## TASOR interaction with DCAF1 in the absence of Vpx

DCAF1/VprBP has been mainly studied as a component of an E3 ubiquitin ligase machinery playing a role in various cellular processes [54]. We showed that DCAF1 does not regulate

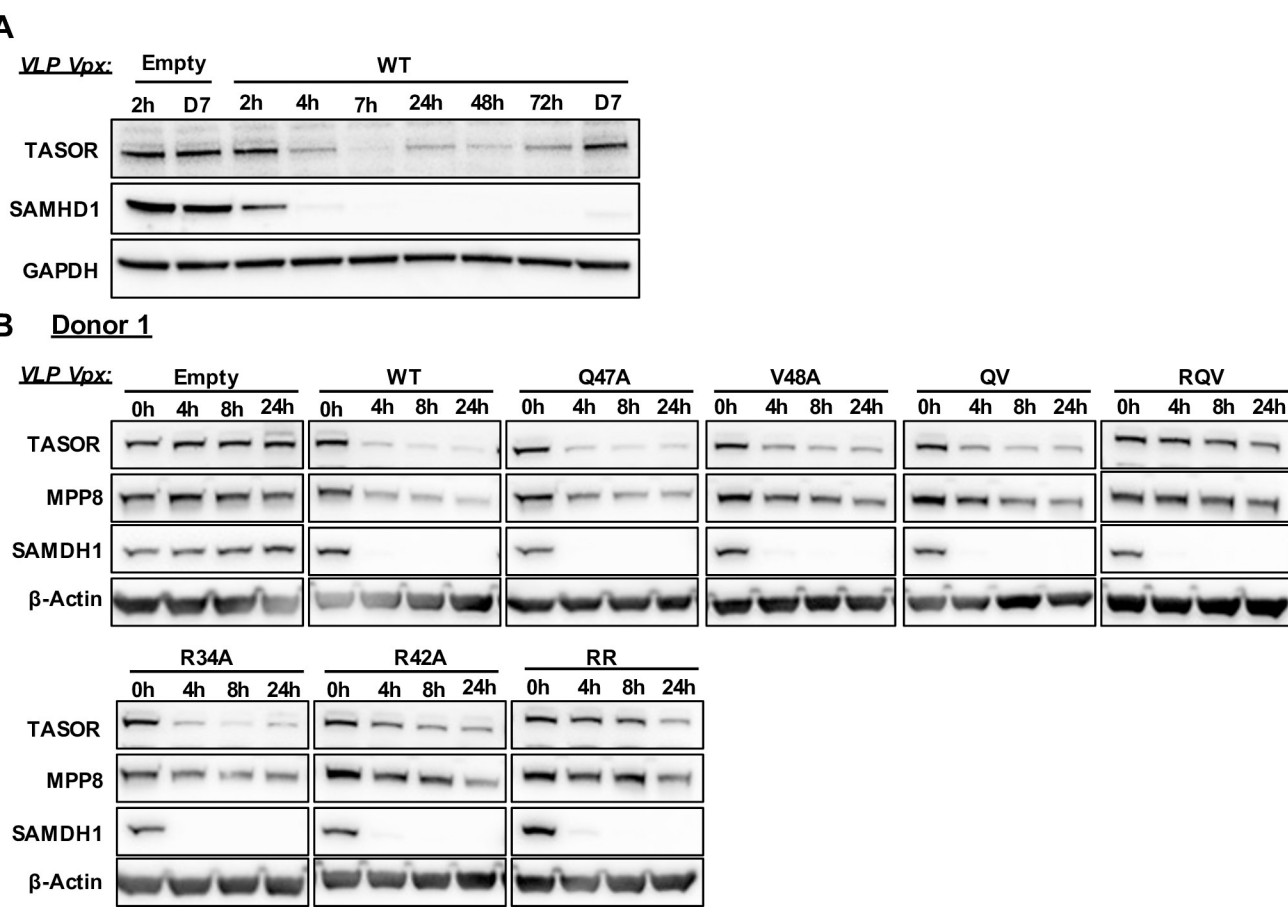

**Fig 6. Vpx R34A-R42A and Vpx R42A-Q47A-V48A mutants are impaired in inducing TASOR, but not SAMHD1, degradation in macrophages.** Long (**A**) and short (**B**) kinetics of TASOR and SAMHD1 degradations by HIV-2.Gh1 Vpx WT or mutants brought by VLP in Monocyte-derived-Macrophages (MDM). Purified monocytes from an healthy donor were differenced 7 days with GM-CSF and M-CSF. After differentiation, MDM were transduced with indicated Vpx-containing VLPs and harvested at indicated times. Whole cell extracts were analyzed by western-blot. *QV*; Vpx Q47A-V48A double mutant. *RQV*: Vpx R42A-Q47A-V48A triple mutant. *RR*: Vpx R34A-R42A double mutant. For this figure, TASOR and SAMHD1 degradation were tested in tree independent donors (*see S7 Fig.*).

endogenous TASOR levels in asynchronized cells (S2B Fig). However, it could be that DCAF1 controls TASOR turnover in a specific cellular context, for instance in a specific window along cell cycle progression or upon DNA damage. Further investigations are required to determine whether DCAF1 could serve as an ubiquitin ligase adaptor for TASOR degradation in physiological conditions.

Interestingly DCAF1 has been shown to negatively regulate transcription and to help the formation of repressive chromatin by binding histone H3 tails protruding from nucleosomes [55]. Moreover, DCAF1 was shown to possess an intrinsic protein kinase activity and is capable of phosphorylating histone H2A on threonine 120 (H2AT120p) in a nucleosomal context [56]. A role of DCAF1 in gene expression has also been uncovered with the discovery of DCAF1 working in conjunction with the Enhancer of Zeste homolog EZH2, a histone methyl transferase associated with transcriptional repression [51]. Therefore, it would be interesting to test the possibility that TASOR works with DCAF1 to repress gene expression. Of note, we found some of the genes regulated by DCAF1 in Kim *et al.* [56] to be upregulated following TASOR depletion in an RNA-seq analysis (unpublished results), supporting the idea of a possible repressive activity of TASOR and DCAF1 on common genes. One may also wonder

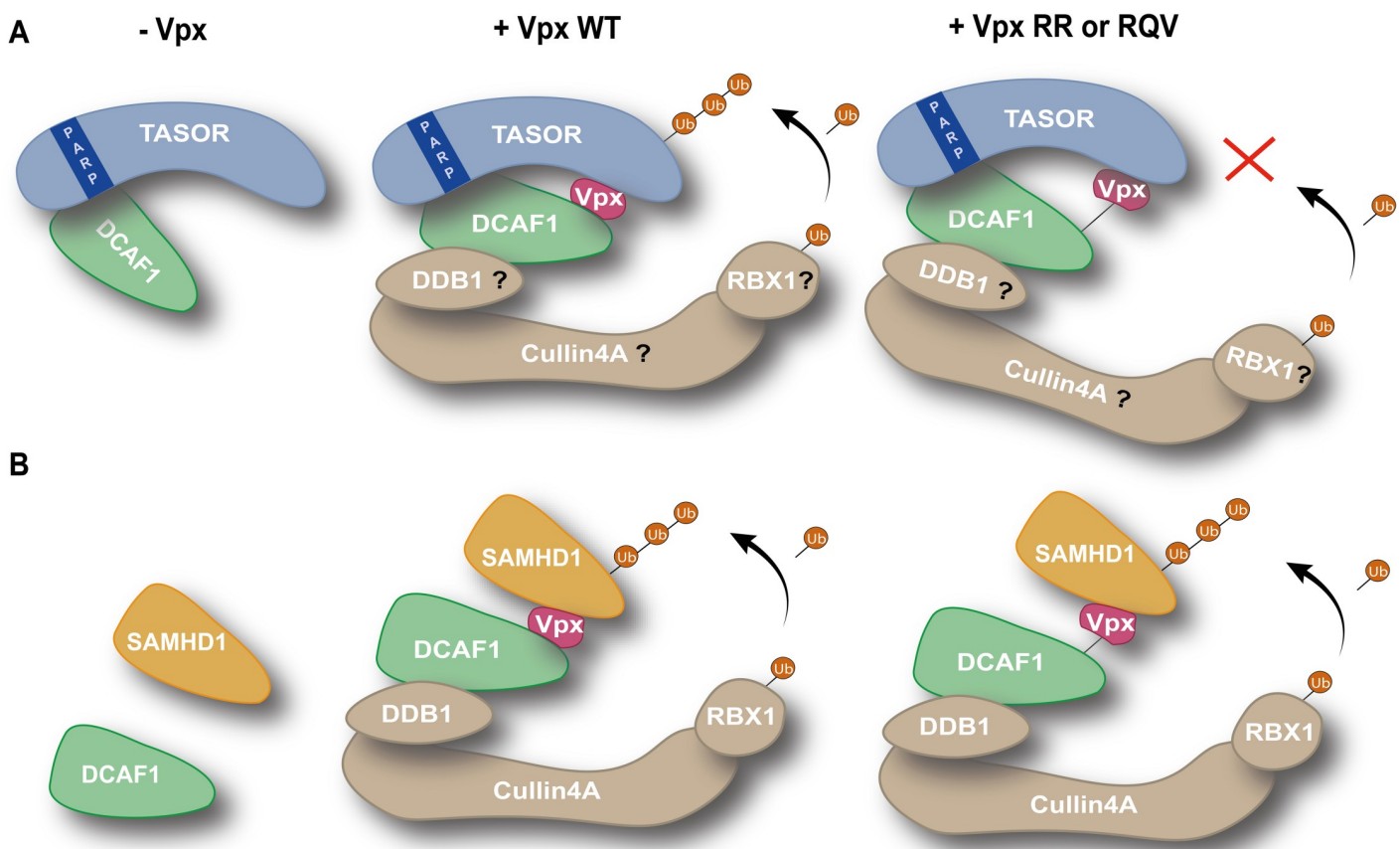

**Fig 7. Working model.** (**A**) TASOR antagonism. In the absence of Vpx, TASOR can be found in association with DCAF1. TASOR PARP-*like* domain is involved in this interaction. By co-binding to TASOR and to DCAF1, Vpx WT would create another indirect contacting point between TASOR and DCAF1, increasing the interaction affinity between TASOR and DCAF1. Structural conformation changes would trigger the degradation of TASOR by the whole ubiquitin ligase complex and TASOR poly-ubiquitination. Vpx RR or Vpx RQV mutants bind TASOR but present a lower affinity for DCAF1, which is represented by a black line; in turn, the interaction affinity between TASOR and DCAF1 is not sufficient to orientate the whole ubiquitin ligase complex in order to trigger TASOR degradation. Of note, whether TASOR-DCAF1 and TASOR-Vpx interactions are direct or not is unknown. (**B**) SAMHD1 antagonism. SAMDH1 does not interact with DCAF1 in the absence of Vpx. Vpx bridges the two cellular proteins, allowing the recruitment of the whole ubiquitin ligase complex, SAMHD1 ubiquitination and degradation. Vpx RR or RQV are both able to induce SAMHD1 degradation, despite their apparent low binding affinity for DCAF1.

whether DCAF1, together with TASOR, could regulate HIV transcription. While DCAF1 is well-known as the ubiquitin ligase adaptor hijacked by Vpr and Vpx, a direct role of DCAF1 in viral transcription has not been investigated yet. In addition to a potential role in transcriptional repression, a cooperation between TASOR and DCAF1 might be at stake in the response to DNA damage. Indeed, DCAF1 interacts with Damage specific DNA Binding protein 1 (DDB1), which is found in complex with the PARP-domain containing PARP1 protein, a sensor of DNA damage (reviewed in [57]). The HUSH complex playing an important role in the epigenetic repression of integrated HIV and recently integrated retroelements, it would be of interest to investigate a role of TASOR in controlling expression of genes near DNA breaks.

### Vpx-mediated TASOR degradation: Binding to HUSH

We were able to highlight several positions in Vpx that are important for HUSH, but not for SAMHD1 antagonism, mainly in α-helices 1 and 2 and in the C-terminal tail of the viral protein. The integrity of several of these residues appears to be important for DCAF1 binding,

whereas none of them seem required for TASOR binding. This result, together with the ability of TASOR to interact with DCAF1, led us to question whether the interaction of Vpx with DCAF1 could by itself trigger HUSH degradation. However, we do not favor this hypothesis because the interaction between Vpx and TASOR does not depend on DCAF1. Indeed, on the one hand, the Vpx Q76R mutant deficient in binding to DCAF1 is able to interact with TASOR and on the other hand, Vpx interacts with TASOR even when DCAF1 is depleted. Altogether, the Vpx viral determinants involved in HUSH binding remain to be discovered. Furthermore, whether this interaction is direct needs to be investigated.

## Vpx-mediated TASOR degradation: Binding to DCAF1

Our results show that DCAF1 is better immunoprecipitated with TASOR in the presence of Vpx. Is it that Vpx induces a conformational rearrangement of the TASOR-DCAF1 complex allowing TASOR to better interact with DCAF1? Does Vpx create new contact points between a preexisting DCAF1-TASOR complex, modifying TASOR/DCAF1 complex conformation allowing TASOR targeting by the ubiquitination complex? Does Vpx bring more DCAF1 in the vicinity of TASOR? In other words, we do not yet know whether Vpx uses the DCAF1 molecule already in association with TASOR or whether Vpx reprograms a new DCAF1 molecule in order to promote TASOR degradation. Nonetheless, we rather favor the hypothesis that Vpx helps TASOR to interact better with DCAF1, since the interaction between TASOR-ΔPARP and DCAF1 is no more stabilized in the presence of Vpx, while TASOR-ΔPARP binds properly to Vpx (model Fig 7). In turn, we speculate that a conformational change could lead to efficient ubiquitination of TASOR, and thus to its efficient degradation by the proteasome. Strikingly, several Vpx mutants impaired for HUSH antagonism had a defect in DCAF1 binding, while none of them were impaired in TASOR binding. Namely, Vpx R42A, Vpx RR and Vpx V48A show reduced binding affinity to DCAF1, but not to TASOR, while being impaired in TASOR degradation in the T-cell line. Vpx R34A presents a moderate phenotype, with reduced affinity for DCAF1 like Vpx R42A, but not as defective in TASOR degradation. Consistently, Vpx R42A and Vpx RR do not stabilize the interaction between DCAF1 and TASOR, whereas Vpx R34A can still do so. Thus, determinants other than DCAF1 binding could be at stake to explain loss of HUSH antagonism, such as a suitable gap between TASOR, DCAF1-associated E3 ubiquitin ligase and the E2 ubiquitin transferase enzyme. Alternatively, unknown components could contribute to Vpx-mediated degradation of TASOR. Further structural analysis by Cryo-Electro Microscopy of the ubiquitin ligase complex would be necessary to understand the positioning of TASOR and DCAF1.

Importantly, the phenotype of Vpx mutants is slightly different in macrophages, in which Vpx R34A, Vpx R42A and Vpx V48A still induce TASOR degradation. Only Vpx R34A-R42A and Vpx RQV mutants, which are strongly impaired in DCAF1 binding, do not efficiently degrade TASOR in macrophages. We wonder whether this could result from a lower expression of TASOR in macrophages compared to T cells, with Vpx being the limiting factor in T cells to remove all DCAF1-bound TASOR, in line with a stoechiometric mechanism. In contrast, SAMHD1 is perfectly degraded in macrophages irrespective of whether Vpx binds efficiently or not DCAF1, in line with a catalytic mechanism. Alternatively, the different abilities of some Vpx mutants to induce TASOR degradation in macrophages, but not in T cells, may also rely on the need or use of cell-specific host factors. The ability of some Vpx mutant to induce SAMHD1 degradation, while binding to DCAF1 was severely impaired, was also very intriguing. This reminds us of SIVdeb Vpx ability to induce SAMHD1 degradation without apparent binding to DCAF1 [58]. Whether an alternative ubiquitin ligase could be used by Vpx to induce SAMHD1 degradation in some specific circumstances remains a possibility.

## Studying the impact of HIV infection in myeloid cells

The use of Vpx mutants capable of degrading SAMHD1, but not HUSH, could be useful for future studies to investigate the impact of HIV infection in myeloid lineages. Indeed, to date, several studies have used Vpx to overcome the reverse transcription blockage in macrophages or dendritic cells to efficiently infect these cells and study the impact of the infection on the cellular landscape [59,60]. One may wonder whether the reported effects are actually the result of the infection or whether they could also result from HUSH degradation. This could be particularly true when studying the modification of the chromatin environment or HIV-induced innate sensing, as HUSH could interfere with both pathways. Indeed, it has been proposed that the regulation of LINE-1 by HUSH serves as a gatekeeper of type 1 interferon signaling, which, when deregulated, could lead to autoinflammatory diseases [61]. On the other hand, we have seen that TASOR protein levels return rapidly after degradation by Vpx, which may reduce the side effects of HUSH depletion. Interestingly, SAMHD1 depletion by Vpx is long-lasting in macrophages compared to HUSH depletion. This difference might reflect the different outcomes resulting from the antagonism of these two restriction factors.

## Restriction of retroviruses along evolution

The restriction of retroviruses by host proteins underlines the long co-evolution history between hosts and viruses. On the viral side, residues involved in the binding of the same substrate often differ between different lentiviral lineages, consistent with the molecular arms-race between hosts and viruses. In future studies, we will question whether differences in DCAF1 binding could also be demonstrated between viral proteins from different lineages, which could have some impact on HUSH, but not SAMHD1, antagonism. Understanding virus-host interaction mechanisms is important to better understand viral pathogenesis and to propose therapeutic strategies that could target restriction factors. Exploiting the activity of HUSH could be advantageous in different strategies, depending on whether the objective is to enhance or lock virus expression.

# Material and methods

## Plasmids

Vpx HIV-2.Gh1 and Vpx SIVsmm WT or mutants, tagged with a HA epitope at the N-terminus, are expressed from the pAS1b vector (pAS1b-HA) [11]. All Vpx mutants were produced by site-directed mutagenesis according to Phusion polymerase manufacture guide (Thermofisher), using the pAS1b-HA-Vpx HIV-2.Gh1 WT (UniP18045) as template. The pAS1b-HA Vpx ΔC-ter has been constructed by introducing a stop codon at position 101 by site-directed mutagenesis. Flag-Vpx-HIV-2.Gh1 WT or mutants with a 3xFlag epitope at the N-terminus, are expressed from p3xFlag vector. The TASOR expression vector, pLenti-TASOR-Flag, and the corresponding empty pLenti-Flag vector were purchased from Origene. TASOR-ΔPARP construct was obtained by PCR-mediated removing of the nucleotides corresponding to the amino-acids 106–319. HA-tagged TASOR are expressed from the pAS1b vector. pLenti-TASOR-Flag and pAS1b-HA-TASOR (1–1512) express a TASOR short isoform of 1512 amino-acids (NCBI Reference Sequence: NP_001106207.1) with a Flag epitope at the C-terminus or a HA epitope at the N-terminus. pAS1b-HA-TASOR (1–1670) expresses a TASOR long isoform of 1670 amino-acids (NCBI Reference Sequence: NP_001352564.1). pAS1b-HA-TAROR (1–931) or pAS1b-HA-TASOR (630–1512) have been constructed by InFusion technology (Takara) according to the kit manufacture guide, using HA-TASOR (1–1512) as template. SAMDH1-Flag is expressed from pCDNA3. DCAF1 isoform 1 (UniP Q9Y4B6-1) or

isoform 3 (UniP Q9Y4B6-3), fused with a Myc epitope at the N-terminus, are expressed from the pCS2 vector. Amino acids 225 to 673 present in isoform 1 are absent from isoform 3. Vpr SIVagm.ver9063 with an HA-epitope at the N-terminus, is expressed from the pAS1b vector.

## Cell culture

Cell lines were regularly tested for mycoplasma contamination: contaminated cells were discarded to perform experiments only with non-contaminated cells. Cells were cultured in media from GIBCO: DMEM (HeLa, 293FT) or RPMI (THP-1, J-Lat), containing 10% heat-inactivated fetal bovine serum (FBS, Eurobio), 1,000 units ml$^{-1}$ penicillin, 1,000 µ g ml$^{-1}$ strep-tomycin and 2 mM glutamine (RPMI only) (Life Technologies). Cells were checked perma-nently according to morphology and functional features (SAMHD1 expression for THP-1 cells; no adherence and low GFP expression for J-Lat A1 T cells, morphology for 293FT and HeLa cells). 293FT cells, optimized from VLP production, were a gift from N. Manel. J-Lat A1 T cells were a gift from E. Verdin. All overexpression experiments were performed with the help of CaCl2 and HBSS (51558-Merck)

## siRNA treatment

siRNA transfections were performed with DharmaFECT1 (Dharmacon, GE Lifesciences). The final concentration for all siRNA was 40nM. The following siRNA was used: siDCAF1: GGAGGGAAUUGUCGAGAAU (Dharmacon). The non-targeting control siRNAs (MIS-SION siRNA Universal Negative Control #1, SIC001) were purchased from Sigma Aldrich.

## Isolation of primary cells

PBMCs from the blood of anonymous donors (obtained in accordance with the ethical guidelines of the Institut Cochin, Paris and *Etablissement Français du Sang*) were isolated by Ficoll (GE Healthcare) density-gradient separation. Monocytes were isolated by positive selection with CD14 magnetic MicroBeads (Miltenyi Biotec). Monocyte-derived macrophages (MDMs) were obtained by 7 days stimulation with 20 ng.ml$^{-1}$ macrophage colony-stimulating factor (M-CSF) and 10 ng.ml$^{-1}$ granulocyte-macrophage colony-stimulating factor (GM-CSF) (Miltenyi Biotec).

## Virus-*Like* Particle (VLP) production and transduction

VLPs were produced in 293FT cells by cotransfection of envelope and packaging vectors by the calcium-phosphate precipitation method. 3.10$^6$ cells were plated the day prior transfection in 10 cm culture dishes. 3µg of VSV-G plasmid, 8µg of SIV3+ ΔVprΔVpx packaging vector (a gift from N. Landau described in [62]) and 8µg of pAS1B-HA-Vpx (WT or mutants) or pAS1b-HA (for empty VLP) or pAS1b-HA-VprSIVagm.ver9063 were then transfected. Cell culture medium was collected 72h after transfection and filtered through 0.45 µm pore filters. VLPs were concentrated by sucrose gradient and ultracentrifugation (1h30 at 100 000*g*). The quality of VLP production and Vpx incorporation was analyzed and quantified by revelation of HA-Vpx and HIV-2 capsid (p27) levels by Western blot. VLP volumes were adjusted in order to transduce the same quantity of Vpx WT and mutants onto cells. J-Lat A1 suspension cells were transduced with VLPs 6h in reduced medium prior to mock or overnight TNF-α (1 ng ml$^{-1}$) treatment.

## Flow cytometry analysis

Cells were collected and resuspended in PBS-EDTA (0.5mM). Data were collected and ana-lyzed with a BD Accuri C6 cytometer and software v100.264.21. At least 10,000 events in P1

were collected, the GFP-positive population was determined using untreated J-Lat A1 T cells according to the low percentage of GFP expressing cells. The same gate was maintained for all conditions. Analysis was performed on the whole GFP-positive population.

## Immunoprecipitation and western blot

For HA-Vpx (WT or mutant), Flag-Vpx, TASOR-Flag (WT or ΔPARP), HA-TASOR, and SAMHD1-Flag immunoprecipitations: HeLa cells grown in 10 cm dishes were co-transfected by the calcium-phosphate precipitation method with pAS1b-HA or pAS1b-HA-Vpx (WT or mutant) or pELR65-SBP-Flag or pELR65-SBP-Flag-Vpx and pLenti-Flag or pLenti-TASOR--Flag (WT or ΔPARP) or pAS1B-HA-TASOR (constructions) or pcDNA3-Flag or pcDNA3--SAMDH1-Flag. 48h post-transfection, cells were treated or not with 10μM of proteasome inhibitor ALLN (CAS 110044-82-1, Santa Cruz) for 5h then cells were lysed in 700μL of RIPA buffer (50mM Tris-HCl pH7.5, 150mM NaCl, 10% Glycerol, 2mM EDTA, 0.5% NP40) containing an anti-protease cocktail (A32965, ThermoFischer). Cell lysates were clarified by centrifugation (10min, 12,000*g*) and 500 μg of lysate was incubated with pre-washed EZview Red ANTI-HA Affinity Gel (E6779, Merck) or ANTI-FlagM2 Affinity Gel (A2220, Merck) at 4˚C, under overnight rotation. After three washes in wash buffer (50mM Tris-HCl pH7.5, 150mM NaCl), immunocomplexes were eluted with Laemmli buffer 1X with 20mM DTT and were separated by SDS-PAGE (Bolt Bis-Tris, 4–12%, Life Technologies). Following transfer onto PVDF membranes, proteins were revealed by immunoblot. Signal were acquired with Fusion FX (Vilber Lourmat) and for further analysis using Fusion software and Image J. The following antibodies, with their respective dilution in 5% skimmed milk in PBS-Tween 0.1%, were used: anti-HA-HRP (3F10) (N˚12013819001, Roche) 1/10,000; anti-FLAG-HRP (A-8592, lot 61K9220, Sigma) 1/10,000; anti-HA (HA-7, H3663, lot 066M4837V, Merck) 1/1,000; anti-Flag M2 (F1804-200UG- lot SLCD3990, Merck) 1/1,000; anti-TASOR (HPA006735, lots A106822, C119001, Merck) 1/1,000; anti-MPP8 (HPA040035, lot R38302, Merck) 1/1,000; anti-βActin (AC40, A3853, Merck) 1/1000; anti-αTubulin (T9026-.2mL, lot 081M4861, Merck) 1/1,000; anti-GAPDH (6C5, SC- 32233, Santa Cruz) 1/1,000; Anti-Vpx has been provided by the NIH AIDS research and reference reagent program (ref EVA3073) 1/500. All HRP-conjugated secondary antibodies, anti-mouse (31430, lot VF297958, ThermoFisher) and anti-rabbit (31460, lots VC297287, UK293475 ThermoFisher), were used at a 1/20,000 dilution before reaction with Immobilon Classico (WBLUC0500, Merck Millipore) or Forte (WBLUF0100, Merck Millipore) Western HRP substrate.

## 3D structural analysis

The (C-ter DCAF1/SIVsm Vpx/C-ter SAMHD1) ternary complex structure was obtained from PDB 4CC9 based on[42]. Structure analysis was performed with Pymol Software (Python).

## Supporting information

**S1 Fig. Interaction between Flag-Vpx, unable to form an apparent dimer, and TASOR. (A)** The 30-kDa form of Vpx is non-denaturable in RIPA or 8M Urea lysis buffer. Increasing quantities of plasmid encoding HA-Vpx WT were expressed in HeLa cells and then the cells were lysed by RIPA lysis buffer or 8M Urea lysis buffer (*composition*: 8M Urea; 2M Thiourea; 4% CHAPS; 30mM Tris HCl pH 8,5). **(B)** The 30-kDa form of Vpx is revealed by an anti-Vpx antibody. HA-Vpx HIV-2.Gh1 or HA-Vpx SIVsmm were expressed in HeLa cells. The whole cells extract was analyzed by Western-blot, first by using an-anti-Vpx antibody, and then with an anti-HA antibody. **(C)** Flag-Vpx WT from HIV-2.Gh1 interacts with HA-TASOR. Flag-Vpx

WT was co-expressed with HA-TASOR short isoform in HeLa cells, then an anti-Flag immunoprecipitation was performed.
(TIF)

**S2 Fig. Interactions between TASOR and DCAF1 and between TASOR and Vpx.** (**A**) TASOR interacts with the two isoforms of DCAF1 in an overexpression system. HA-TASOR short isoform was co-expressed with Myc-DCAF1 isoform 1 (iso1) or isoform 3 (iso3) in HeLa cells, then an anti-HA immunoprecipitation was performed. (*) The Myc-DCAF1 band is detected in the HA-TASOR panel. (**B**) Endogenous TASOR protein level is not affected following DCAF1 depletion. HeLa cells were transfected with 40nM of siCTL (-) or siDCAF1 (+) and cells were harvested at 48h and 72h. (**C**) Flag-Vpx interacts with HA-TASOR in absence of DCAF1. HeLa cells were treated with siRNA CTL or siRNA DCAF1. After 24h, Flag-Vpx WT was co-expressed with HA-TASOR for 48h, then an anti-Flag immunoprecipitation was performed. In each panel, the indicated proteins were revealed by western blot.
(TIF)

**S3 Fig. Analysis of HA-Vpx (WT or mutants) incorporation into VLP by western-blot.** For each panel, VLP were produced in 293FT by co-transfection of a packaging vector, an envelope VSVg vector and a vector encoding HA-Vpx (WT or mutants). 72h post transfection, supernatants were harvested and the VLPs were concentrated by ultracentrifugation. 12 μL of each were analyzed by western blot. VLP production was checked with anti-P27 (HIV-2 capsid) antibody and HA-Vpx incorporation with an anti-HA antibody. (**A**) Western blot of VLP incorporation for Figs 3A, 3B, 3C, 4B and 4C. (**B**) Western Blot of VLP incorporation for Fig 5D, 5E and 5F. (**C**) Western Blot of VLP incorporation for Fig 6B.
(TIF)

**S4 Fig. The inability of a set of SIVsmm or HIV-2 Vpx mutants to degrade TASOR correlates with a lower binding affinity to DCAF1.** (**A**) SIVsmm Vpx R42A, V48A and QV mutants show a lower binding affinity to DCAF1. HA-Vpx WT from SIVsmm or indicated mutants were expressed in HeLa cells, then an anti-HA immunoprecipitation was performed. (**B**) TASOR degradation phenotype of Flag-Vpx HIV-2.Gh1 WT or mutants. *Left*. Analysis of Flag-Vpx (WT and mutants) incorporation into VLP by western-blot. *Right*. J-Lat A1 T cells were treated with VLPs containing Flag-Vpx WT or indicated mutants. After overnight treatment, the whole cell extracts were analyzed by western blot. The immunoblot is representative of 3 independent experiments. (**C and D**) Flag-tagged Vpx R34A, R42A, RR and QV mutants are less affine for DCAF1 (**C**) but not for TASOR (**D**). Flag- HIV-2.Gh1 Vpx WT or indicated mutants were expressed in HeLa cells, then an anti-Flag immunoprecipitation was performed. The shown Immunoblots are representative of 2 independent experiments.
(TIF)

**S5 Fig. TASOR still interacts with Vpx depleted from its C-terminal poly-proline tail.**
HA-Vpx WT or HA-ΔC-ter Vpx and TASOR-Flag were co-expressed in HeLa cells, then an anti-HA immunoprecipitation was performed.
(TIF)

**S6 Fig. Some Vpx mutants defective for HUSH antagonism interact with TASOR.** (**A**) HA-Vpx WT or indicated mutants and TASOR-Flag were co-expressed in HeLa cells, then an anti-HA immunoprecipitation was performed. (**B**) TASOR-Flag and HA-Vpx WT or indicated mutants were co-expressed in HeLa cells, then an anti-Flag immunoprecipitation was performed.
(TIF)

**S7 Fig. TASOR and SAMHD1 degradations induced by Vpx WT or mutants in macrophages from additional donors.** (**A**) Short Kinetics of TASOR and SAMHD1 degradations by HIV-2.Gh1 Vpx WT or mutants delivered to Monocyte-derived-Macrophages (MDM) by VLP.Purified monocytes from healthy donors were differentiated 7 days with GM-CSF and M-CSF. After differentiation, MDM were transduced with indicated Vpx-containing VLPs and harvested at indicated times. Whole-cell extracts were analyzed by western-blot. *QV*; Vpx Q47A-V48A double mutant. *RQV*: Vpx R42A-Q47A-V48A triple mutant. *RR*: Vpx R34A-R42A double mutant. (**B**) Short kinetic of TASOR degradation by SIVagm.ver9063 Vpr in MDM. SIVagm.ver Vpr is unable to induce human SAMDH1 degradation.
(TIF)

**S1 Data. Excel spreadsheet containing, in separate sheets, the underlying numerical data for Fig panels 3B, 4B, 5B and 5D.**
(ZIP)

## Acknowledgments

We thank Claudine Pique and all members of the RIL team for fruitful comments during lab meetings for the project progress. We also thank Claudine Pique for her precious reading of the manuscript. We thank Lucie Etienne and Andrea Cimarelli for discussions on the concepts of the project and for their comments on a previous version of the manuscript. We thank Ghina Chougui for her preliminary analysis of some Vpx mutants. The authors also thank E. Verdin for J-Lat clones and N. Landau for SIV packaging constructs. The authors acknowledge the Cytometry and Immunobiology Facility of the Cochin Institute.

## Author Contributions

**Conceptualization:** Michaël M. Martin, Roy Matkovic, Pauline Larrous, Marina Morel, Florence Margottin-Goguet.

**Formal analysis:** Michaël M. Martin, Roy Matkovic, Florence Margottin-Goguet.

**Funding acquisition:** Florence Margottin-Goguet.

**Investigation:** Michaël M. Martin, Roy Matkovic, Pauline Larrous, Marina Morel, Angélique Lasserre, Virginie Vauthier, Florence Margottin-Goguet.

**Methodology:** Michaël M. Martin, Roy Matkovic, Florence Margottin-Goguet.

**Project administration:** Michaël M. Martin, Roy Matkovic, Florence Margottin-Goguet.

**Resources:** Florence Margottin-Goguet.

**Supervision:** Florence Margottin-Goguet.

**Validation:** Michaël M. Martin, Roy Matkovic, Pauline Larrous, Marina Morel, Angélique Lasserre, Virginie Vauthier, Florence Margottin-Goguet.

**Visualization:** Michaël M. Martin, Roy Matkovic, Florence Margottin-Goguet.

**Writing – original draft:** Florence Margottin-Goguet.

**Writing – review & editing:** Michaël M. Martin, Roy Matkovic, Florence Margottin-Goguet.

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
