## [Decision Letter · Decision Letter 0]

31 May 2021

Dear Dr. Margottin-Goguet,

Thank you very much for submitting your manuscript "Binding to DCAF1 distinguishes TASOR and SAMHD1 degradation by HIV-2 Vpx" for consideration at PLOS Pathogens. As with all papers reviewed by the journal, your manuscript was reviewed by members of the editorial board and by several independent reviewers. In light of the reviews (below this email), we would like to invite the resubmission of a significantly-revised version that takes into account the reviewers' comments. Please pay particular attention to the suggestions of reviewers 1 and 3 regarding the higher molecular weight/dimers, as well as the minor comments of all three reviewers.

We cannot make any decision about publication until we have seen the revised manuscript and your response to the reviewers' comments. Your revised manuscript is also likely to be sent to reviewers for further evaluation.

Sincerely,

Welkin E. Johnson

Associate Editor

PLOS Pathogens

Susan Ross

Section Editor

PLOS Pathogens

Kasturi Haldar

Editor-in-Chief

PLOS Pathogens

orcid.org/0000-0001-5065-158X

Michael Malim

Editor-in-Chief

PLOS Pathogens

orcid.org/0000-0002-7699-2064

Reviewer's Responses to Questions

**Part I - Summary**

Reviewer #1: Within their manuscript “Binding to DCAF1 distinguishes TASOR and SAMHD1 degradation by HIV-2 Vpx”, Martin and colleagues characterize the interaction of HIV-2 Vpx with the component of the HUSH complex TASOR in great detail on a molecular level.

The manuscript is nicely written and the experiments are technically sound and well controlled. The study is of some interest to the field as it adds new molecular details on the nature of the Vpx-mediated degradation of the HUSH complex/TASOR.

Within a series of convincing immunoprecipitation and western blot experiments Martin et al found that TASOR binds to the E3-Ligase adaptor molecule DCAF1, even in the absence of Vpx. However, HIV-2 Vpx binding to TASOR and DCAF1 stabilizes this interaction. In addition, the authors found that the N-terminal domain of TASOR is important for Vpx interaction and were able to identify patches in Vpx crucial for TASOR inactivation and promoting HIV mini genome expression. Interestingly, the authors also found that the residues in Vpx important for binding to TASOR differs from the ones important for SAMHD1 interaction.

Reviewer #2: The manuscript by Martin and Matkovic et al. describes a rigorous and thorough structure-function analysis of HIV-2 Vpx interaction with known partners DCAF1, TASOR and SAMHD1. Whereas the determinants of Vpx accessory protein from HIV-2/SIVsm have been quite well studied regarding SAMHD1 antagonism, much remained to be understood in the context of HUSH complex degradation. In this context, the authors did an extensive mutagenesis analysis in order to characterize Vpx interactions with DCAF1, TASOR and SAMHD1. Whereas DCAF1 interaction with SAMHD1 depends on Vpx, the authors report here that DCAF1 was able to interact with TASOR in the absence of Vpx, but the latter did increase/stabilize the interaction. Then the authors showed the importance of the PARP-like domain of TASOR in DCAF1 binding; of note this domain was not important for Vpx binding, suggesting that DCAF1 and Vpx bind to different regions of TASOR. Moreover, the authors interestingly identified a series of Vpx mutants unable to induce TASOR degradation while remaining active against SAMHD1. Surprisingly, some of these mutants had a poor binding affinity for DCAF1 but not for TASOR, which highlighted the essential role of DCAF1 in Vpx-mediated degradation of TASOR. In addition, this raised the question of whether little amounts of DCAF1 could be sufficient for Vpx-mediated degradation of SAMHD1, or if an alternative partner could be used in this context. As expected, there was a strict correlation between the ability of Vpx mutants to degrade TASOR and to enhance HIV-1 minigenome expression in a J-lat A1 T cell model. Finally, the authors globally confirmed their observations in primary monocyte-derived-macrophages. Of note, some of the Vpx mutants described here will be particularly useful to specifically study HUSH restriction in primary myeloid cells, where SAMHD1 is also active. The manuscript is very well written, the data are clear and convincing, the findings are highly interesting and they will certainly be of great interest for people in the field. There are only a few minor points to address.

Reviewer #3: The manuscript authored by Martin and coworkers expands on previous observations by the Margottin-Goguet lab relating to the ability of Vpx to degrade the HUSH complex. In this work, Martin et al. provide ample mechanistic details about this relationship and compare it with the Vpx-SAMHD1 relationship. The main conclusions of this work are (1) TASOR interacts with DCAF-1 independently of Vpx (unlike SAMHD1); 2) mutants are presented, which dissect SAMHD1 versus TASOR degradation; and 3) transactivation of the viral promoter segregates with those mutants as one might expect based on HUSH degradation. This is clearly very important work and the data looks convincing. ¬¬Therefore, I only have minor suggestions regarding this work.

The first suggestion is regarding the higher MW band that is thought to be a dimer of Vpx. It is difficult for the reviewer to judge whether this is a dimer because there are no MW markers on any the gels. Secondly, to corroborate that the dimer is indeed Vpx, I would suggest using a urea-PAGE electrophoresis setup, which most likely would reduce Vpx dimers to monomers.

My second suggestion is to demonstrate that TASOR degradation occurs in primary cells (lymphocytes and/or macrophages); is this something that was done in earlier work? If so I am not aware of it but it would be important.

**Part II – Major Issues: Key Experiments Required for Acceptance**

Reviewer #1: Vpx dimerization. The authors use HIV-2 Vpx from Ghana I (line121) and found that HA-tagged WT Vpx form “non-denaturable” dimers. Intriguingly, all Vpx mutants deficient for TASOR binding (Q47A, V48A, and QV) in Figure 3D are monomers.

a) Is the difference in molecular weight between HA-Vpx wt and the mutants important for binding/function? The authors should repeat the experiments with FLAG tagged Vpx wt and mutant proteins (which does not form dimers, Figure S1?)

b) How do the authors know these bands are dimers? Isn’t it possible that Vpx under these conditions is posttranslationally modified or binds to an unknown cellular cofactor? Can the authors find similar dimers (wt) and loss of function and dimerization (mutants) in other HIV-2 or SIV Vpx proteins?

c) The “dimerization” of Vpx within VLPs appears to be less pronounced than in cellular lysates, especially for WT Vpx (compare S2A and S2C to Figure 3)? Why is that the case?

Since the authors do not (cannot ?) convincingly explain the nature of the “dimer band” and given the fact that the very detailed analysis of protein-protein interactions by immunoprecipitation is the most important and major part of the manuscript, the authors need to verify the effects of the Vpx mutants on TASOR and Decaf binding in different Vpx constructs. For example, using at least a core set of FLAG-tagged Vpx mutants (Ghana) or introduce the mutations in other HIV Vpx proteins that do not form the dimers, just to rule out unwanted effects of the “dimer” band on the TASOR/DECAF interaction.

Reviewer #2: (No Response)

Reviewer #3: I have suggested two small experiments in the previous section. Briefly, to demonstrate that the upper band is a dimer of Vpx, and to reproduce degradation of TASOR in primary cells.

**Part III – Minor Issues: Editorial and Data Presentation Modifications**

Reviewer #1: (No Response)

Reviewer #2: Minor issues :

In the figure legends, please specify the number of experimental repeats that were done (e.g. the immunoblots shown are representative of n experiments).

The loading control (B-actin or GAPDH as in the other immunoblots) is missing in Figure 1D.

As mentioned above, an interaction between TASOR and DCAF1 was reported by the authors in the absence of Vpx. The authors state that however that TASOR was not regulated by DCAF1 in the absence of Vpx, as shown by the unchanged levels of TASOR in the absence of DCAF1 (Fig S1C). However, it seems quite clear from Figure S1D and Figure 1D (assuming in the latter case that the total amount of proteins was identical between the different lines, as the loading control is missing), that TASOR seemed expressed to higher levels in the absence of DCAF1. Could the authors please comment this observation? Is this reproducible? Is it due to the ectopic expression of (a tagged version of) TASOR (i.e. HA-TASOR or TASOR-Flag)? (Does the ectopic expression change the localization of a pool of the protein for instance?...).

Fig. 3D: there seemed to be less DCAF1 co-immunoprecipitated with TASOR-Flag in the presence of Vpx QV mutant compared to the condition where Vpx is absent: is it reproducible and significant?

p11 “… macrophages, in which both TASOR and SAMHD1 are present”: either add restriction (SAMHD1 restriction) or replace “present” by “active”, as SAMHD1 is ubiquitously expressed but active in myeloid cells

Reviewer #3: N/A

PLOS authors have the option to publish the peer review history of their article (what does this mean?). If published, this will include your full peer review and any attached files.

Reviewer #1: No

Reviewer #2: No

Reviewer #3: **Yes: **Vicente Planelles
---

## [Editor Report · Decision Letter 1]

15 Oct 2021

Dear Dr. Margottin-Goguet,

We are pleased to inform you that your manuscript 'Binding to DCAF1 distinguishes TASOR and SAMHD1 degradation by HIV-2 Vpx' has been provisionally accepted for publication in PLOS Pathogens.

Best regards,

Welkin E. Johnson

Associate Editor

PLOS Pathogens

Susan Ross

Section Editor

PLOS Pathogens

Kasturi Haldar

Editor-in-Chief

PLOS Pathogens

orcid.org/0000-0001-5065-158X

Michael Malim

Editor-in-Chief

PLOS Pathogens

orcid.org/0000-0002-7699-2064
---

## [Editor Report · Acceptance letter]

21 Oct 2021

Dear Dr. Margottin-Goguet,

We are delighted to inform you that your manuscript, "Binding to DCAF1 distinguishes TASOR and SAMHD1 degradation by HIV-2 Vpx," has been formally accepted for publication in PLOS Pathogens.

Best regards,

Kasturi Haldar

Editor-in-Chief

PLOS Pathogens

orcid.org/0000-0001-5065-158X

Michael Malim

Editor-in-Chief

PLOS Pathogens

orcid.org/0000-0002-7699-2064